# Uncovering Novel Extracellular Matrix Transcriptome Alterations in Lesions of Multiple Sclerosis

**DOI:** 10.3390/ijms25021240

**Published:** 2024-01-19

**Authors:** Erin Laurel Stephenson, Rajiv William Jain, Samira Ghorbani, Rianne Petra Gorter, Charlotte D’Mello, Voon Wee Yong

**Affiliations:** 1Department of Pathology and Laboratory Medicine, University of Calgary, Calgary, AB T2N 2T9, Canada; elstephe@ucalgary.ca; 2Hotchkiss Brain Institute and Department of Clinical Neurosciences, University of Calgary, Calgary, AB T2N 4N1, Canada; rajivwilliam.jain@ucalgary.ca (R.W.J.); samira.ghorbanigazar@ucalgary.ca (S.G.); rianne.gorter@ucalgary.ca (R.P.G.); dmelloc@ucalgary.ca (C.D.)

**Keywords:** multiple sclerosis, extracellular matrix, SPARC, glycoprotein, proteoglycans

## Abstract

The extracellular matrix (ECM) of the central nervous system (CNS) is an interconnected network of proteins and sugars with critical roles in both homeostasis and disease. In neurological diseases, excessive ECM deposition and remodeling impact both injury and repair. CNS lesions of multiple sclerosis (MS), a chronic inflammatory and degenerative disease, cause prominent alterations of the ECM. However, there are a lack of data investigating how the multitude of ECM members change in relation to each other and how this affects the MS disease course. Here, we evaluated ECM changes in MS lesions compared to a control brain using databases generated in-house through spatial mRNA-sequencing and through a public resource of single-nucleus RNA sequencing previously published by Absinta and colleagues. These results underline the importance of publicly available datasets to find new targets of interest, such as the ECM. Both spatial and public datasets demonstrated widespread changes in ECM molecules and their interacting proteins, including alterations to proteoglycans and glycoproteins within MS lesions. Some of the altered ECM members have been described in MS, but other highly upregulated members, including the SPARC family of proteins, have not previously been highlighted. SPARC family members are upregulated in other conditions by reactive astrocytes and may influence immune cell activation and MS disease course. The profound changes to the ECM in MS lesions deserve more scrutiny as they impact neuroinflammation, injury, and repair.

## 1. Introduction

Multiple sclerosis (MS) is a chronic neurodegenerative disease characterized by the recruitment of immune cells into the central nervous system (CNS) and activation of local microglia and astrocytes, associated with the loss of myelin sheaths and oligodendrocytes. Following the destruction of myelin, recovery may occur with the migration of oligodendrocyte precursors to lesion sites and myelin repair. Remyelination, the reformation of myelin, has been a strong focus in the MS field and relies on overcoming inhibitory factors and detrimental neuroinflammation. Less well characterized are the alterations of the extracellular matrix (ECM) in MS lesions.

The ECM is a network of highly interactive components crucial for CNS function. The ECM has distinct components in spatial regions of the CNS and is organized into the basement membrane, interstitial matrix, and perineuronal nets. The major categories of the ECM include glycoproteins, proteoglycans, and collagens. Members of each category contain a variety of functions and structures and are present in several locations. The basement membrane lines parenchymal vessels and primarily consists of type IV collagens, laminins, and heparan sulfate proteoglycans [1]. Perineuronal nets are a network around neuronal bodies and consist of tenascins, hyaluronans, link proteins, and chondroitin sulfate proteoglycans [1]. The interstitial matrix of the CNS consists mainly of hyaluronan molecules and chondroitin sulfate proteoglycans [1]. The ECM serves both structural and signaling roles, including, but not limited to, binding receptors like integrins, maintaining synaptic stability, and forming an extracellular microenvironment that alters cell biology; this has been extensively reviewed previously [1].

There is increasing appreciation that there are global alterations in the composition, density, and architecture of the ECM in MS. Basement membrane components (e.g., laminins, heparin sulfate proteoglycans, fibronectin, hyaluronan, vitronectin, and collagen type IV) are upregulated in lesions and perivascular cuffs and correlate with blood–brain barrier disruption and inflammation [2,3]. At sites of blood–brain barrier disruption, there is an accumulation of fibrinogen which is associated with microglial activation and axonal damage [4]. A study of 50 ECM RNA transcripts found elevated collagens, laminins, and proteoglycans in demyelinated MS lesions [5]. Chondroitin sulfate proteoglycans, particularly the versican V1 isoform, are upregulated in MS lesions and are accumulated in regions of infiltrating immune cells [6,7].

ECM components alter the activation of resident and infiltrating immune cells to influence both inflammatory damage and remyelination outcomes. For example, laminins can influence sites of cellular trafficking [8], and chondroitin sulfate proteoglycans can promote inflammation and trafficking in macrophages [7]. Cellular communication network factor 3 (CCN3), a matricellular growth factor, secreted by regulatory T cells enhances remyelination after lysolecithin or cuprizone demyelination [9]. Hyaluronan is accumulated in MS lesions, and some reports detail that its high-molecular-weight form permits remyelination [10], but, conversely, another study identified low-molecular-weight hyaluronan oligomers as non-permissive for remyelination through TLR2 receptors [11].

Beyond the list of core ECM members (proteoglycans, glycoproteins, and collagens), there is a much larger group of ECM-associated proteins predicted to be affiliated with the ECM; the combination of these constituents is referred to as the “matrisome” [12]. This larger list includes ECM-modifying enzymes, such as matrix metalloproteinases (MMPs), ADAMTs (A Disintegrin and Metalloproteinase with Thrombospondin motifs)*,* and protease inhibitors. MMPs are known to be elevated in MS and have pathological roles [13].

This study provides an in-depth characterization of global ECM changes within chronic active lesions of MS using two databases, one with in-house-generated spatial transcriptome sequencing and the other of a public single-nucleus RNA sequencing database. Our findings illustrate the profound changes the ECM undergoes during MS and provide a resource for further investigations into the ECM as a target to influence the MS disease course.

## 2. Results

### 2.1. Classifying Components of the Extracellular Matrix

We started our investigations by locating a thorough categorization of the ECM. Outside of the core ECM, the list of ECM-associated proteins is difficult to define since ECM protein domains are shared with non-ECM proteins. Gene ontology categories are of limited help, as they contain lengthy and unannotated lists of proteins. Naba et al. (2012) have expertly curated a thorough categorization of ECM based on protein domains using InterPro and SMART databases [12,14]. Their list of core ECM components corresponds with our prior knowledge and expertise of the ECM. Their ECM list used in this study comprised around 300 proteins with approximately 200 “major glycoproteins” (Appendix A), which includes structural components (e.g., laminins) as well as matricellular components (e.g., thrombospondin, SPARC/SPARC-like 1, etc.). Appendix A is a list of proposed glycoproteins that are less well-known but are classified based on common protein domains.

Proteoglycans consist of a protein core with attached glycosaminoglycans chains that comprise most of their mass and provide a high overall negative charge. Proteoglycans can be divided into intracellular, cell surface (transmembrane or GPI-anchored), pericellular/basement membrane, and extracellular [15] (Appendix A). Extracellular proteoglycans include the CNS-enriched hyalectins/lectins, small leucine-rich proteoglycans, and SPOCK proteins (e.g., SPOCK1-3). There are some glycoproteins that can be modified by glycosaminoglycan side changes (e.g., certain collagens, CD44, agrin, etc.) but have been grouped with glycoproteins since the glycosaminoglycan side chains do not represent the majority of their mass. Collagen components were grouped into fibrillar-forming, facit-forming, short-chain collagens, CNS basement membrane-enriched, von Willebrand factor A-domain-containing proteins, and non-classified, and they are listed in Appendix A.

### 2.2. Spatial RNA Transcriptome Highlights Important Spatial Changes in ECM Members

Spatial RNA transcriptome data were generated from non-demented controls (*n* = 2) and people with MS (*n* = 3) using the 10x Genomics Visium platform. Reference tissue sections were stained with myelin-basic protein (yellow), neurofilament-L/M/H (magenta), laminin (red), CD45 (green), and a nuclear DAPI stain (blue) to demarcate the normal-appearing gray matter, normal-appearing white matter, and demyelinated lesions (Figure 1A). Lesions were classified according to the system of Kuhlmann et al. (2017) [16]. Three mixed active/inactive lesions (referred to as chronic active lesions henceforth) were analyzed, two fully within white matter, and one leukocortical lesion. Tissue sections were normalized and mapped using Space Ranger to generate spatially correlated RNA sequencing mapped to H&E tissue sections. Visualization of data was performed using 10x Genomics software Loupe Browser (v6.3.0) which was used to define the borders of periplaque white matter (“periplaque WM”) adjacent to the lesion, active edge, and inactive lesion core of chronic active lesions.

LoupeBrowser generated differentially expressed genes (adjusted *p*-value with Benjamini–Hochberg procedure) for our list of ECM genes. Expression values are reported as median-normalization average of a gene, which represents the mean of observed UMI counts per spot. Using a conservative cut-off of greater than 0.05 median-normalized expression (UMI count per spot) to filter out low levels of ECM members, there were 20 significantly (adjusted *p*-value < 0.05) differentially expressed ECM members across white matter conditions (Figure 1B,C). Figure 1C shows the top 10 differentially expressed genes in control and white matter lesions ranked in order of greatest median-normalized expression in white matter lesion cores. These results highlight that the majority of the differentially expressed genes are in the categories of proteoglycans and glycoproteins.

There was minimal detectable expression of collagens within our dataset. The expression of collagens is limited within the CNS, with collagen type IV clustered around basement membranes. Another group of ECM not detected was laminins. Although laminins are shown to be important in regulating cellular trafficking in MS, their expression is confined to basement membranes and may explain their low detected levels in our dataset.

*SPARC* was the greatest differentially upregulated gene in all MS conditions compared to control white matter (log2 fold change of −2.51 (adjusted *p*-value = 0.018) in control white matter compared to all other MS lesion conditions). *SPARC* was progressively upregulated in periplaque white matter, MS lesion edges, and the greatest expression was in MS lesion core (Figure 1B,C). *SPARC* had the highest normalized mean expression of any of the ECM members in MS lesion core and the second highest mean counts in MS lesion edge (second to *SPP1*) of chronic active lesions.

The SPARC family are matricellular proteins that can be grouped into four groups: (1) SPARC (osteonectin, BM-40) and hevin/SPARC-like 1 (SPARCL1); (2) secreted modular calcium binding protein (SMOC) 1 and 2; (3) SPOCK/testicans 1, 2, and 3; and (4) follistatin like protein 1 (fstl-1, TSC-36/Flik, follistatin related protein (FRP), and TGF-β inducible protein). SPOCK1-3 contain a distinct glycosaminoglycan-binding domain which puts them in the category of proteoglycans. Among the 20 greatest differentially expressed genes in white matter MS lesions (Figure 1B), 4 were members of the SPARC family (i.e., *SPARC* and *SPOCK1*, *2*, and *3*). However, while *SPARC* was upregulated in MS-related conditions, *SPOCK 1* and *3* showed the opposite trend, with decreased log2 fold in MS inactive core compared to other conditions (e.g., *SPOCK1* −1.033 log2 fold change, adjusted *p*-value 0.00223; and *SPOCK3* −1.500 log2 fold change, adjusted *p*-value 0.000196).

*SPP1*, the gene that encodes osteopontin, had the greatest UMI counts of ECM members in MS lesion edges. Osteopontin was downregulated in MS lesion cores compared to all other conditions (due to the higher expression in MS lesion edges and periplaque white matter) but still had the second greatest mean UMI count of ECM members in MS lesion cores (Figure 1B,C). Osteopontin has been identified as an upregulated protein in MS [17].

*VCAN*, the gene encoding versican, was also identified in our analysis as significantly upregulated, with the greatest expression in lesion cores and 1.08 log2 fold change compared to all other categories; this is consistent with previous reports that identified versican upregulation in both an inflammatory demyelinating mouse model and in MS lesions [7].

Due to the low yield of mRNA from the spatial dataset, the transcripts of proteases (MMPs and ADAMS) and other ECM-associated proteins were not present at detectable levels.

### 2.3. The Extracellular Matrix Transcriptome from Single-Nucleus RNA Sequencing of Chronic Active Lesions

There is a newfound focus in MS research to determine important molecular players that are contributing to different types of lesions in MS. Absinta et al. (2021) conducted single-nucleus RNA sequencing to profile different stages of demyelinated white matter lesions [18]. Their dataset is from five progressive MS patients and they sequenced periplaque white matter (*n* = 4), chronic active edges (*n* = 6), chronic inactive edges (*n* = 5), lesion cores (*n* = 2), and normal white matter from neurologically healthy brains (*n* = 3). Their results emphasized the diverse changes in glial and immune cells within lesions. We independently analyzed this publicly available dataset to determine the presence of ECM members in the different lesion types. We performed independent filtering and quality control, and our analysis yielded 99,432 single-nucleus transcriptomic profiles, slightly greater than the 66,432 in the original study due to different inclusion criteria.

A total of 276 genes of the core matrisome were detected in the Absinta dataset. Of these genes, 115 were significantly differentially expressed within one cluster compared to all other clusters, after filtering out for low expression (0.05 normalized median UMI counts). There were greater levels of detectable genes in this dataset than the spatial dataset, which allowed for a greater analysis of the trends in ECM levels. Of these ECMs, the majority were “major glycoproteins”, 28 were in the category of “proposed glycoproteins”, 28 were collagens, and eight were laminin components.

Figure 2A shows the heatmap expression of major glycoproteins, with genes ranked in terms of median-normalized expression in lesion core conditions (top = greatest; bottom = lowest). The trends show a clear upregulation of glycoproteins in chronic active edges, chronic inactive edges, and the lesion core. Appendix A shows the log2 fold changes in the proposed glycoproteins, which showed similar trends to the major glycoproteins. Within the list of 34 proteoglycans, 10 were significantly differentially expressed (Figure 2B). Their trends were similar to glycoproteins, with the majority upregulated in chronic active edges, chronic inactive edges, and lesion cores. *SPP1* again emerged as an upregulated differentially expressed gene within the chronic active edge (Figure 2A).

The changes in the SPARC family showed similar trends to the spatial dataset. *SPARC* was 1.16 log2 fold upregulated in chronic inactive and 0.45 log2 fold in chronic active lesion edges. *SPARCL1* had greater median-normalized average expression in this dataset than *SPARC* and was 0.75 log2 fold upregulated in chronic active and chronic inactive edge conditions compared to all other conditions. *SPOCK1* and *SPOCK3* showed the opposite trend, with log2 fold decreases in active edge, inactive edge, and lesion cores, similar to their decreased expression in lesion cores in the spatial dataset.

Compared to the profound changes in ECM in MS lesion conditions, there were minimal changes to glycoproteins when directly comparing periplaque white matter and the control. Only *CD44* increased in periplaque WM compared to control WM (*CD44* log2 fold change 1.1 for periplaque versus control white matter, adjusted *p*-value 0.0001) and four genes were sufficiently expressed and significantly decreased in periplaque versus control white matter (i.e., *CD74* log2 fold change −0.7, adjusted *p*-value 0.03; *FRAS2* log2 fold change −0.8, adjusted *p*-value 0.01; *THBS2* log2 fold change −1.1, adjusted *p*-value 2.5 × 10^−5^; and *GLDN* log2 fold change −1.3, adjusted *p*-value 1.35 × 10^−8^). These results are consistent with the heatmaps in Figure 2, which show similar trends of control and periplaque conditions.

The heatmap of laminins (Figure 2C) shows the trend of increased log2 fold upregulation in lesion cores, although with low median-normalized expression. Figure 2D shows the top 10 differentially expressed glycoproteins in order of greatest expression in lesion cores.

The distribution of collagens is shown in Appendix A. Genes were ranked for their mean expression in chronic active lesions (top = greatest). As expected, there was low detected expression in fibrillar-forming collagens.

With regard to proteases capable of processing ECM components, several ADAMs (a disintegrin and metalloproteinase), ADAMTSs (ADAMs with thrombospondin motifs), and a smaller number of MMPs (matrix metalloproteinases) were detected (Figure 3). Higher expression tended to be in the lesion core compared to other types of lesions or control specimens.

### 2.4. SPARC and SPARCL1 Are Expressed within Activated Astrocytes

When we assessed the expression of SPARC family members amongst the different cell clusters identified by Absinta et al. (2021) in their dataset [18], we found the SPARC family present in multiple cell clusters. *SPARC* (1.6 log fold, adjusted *p*-value 5.93 × 10^−73^) and *SPARCL1* (1.03 log fold, adjusted *p*-value 1.7 × 10^−21^) were differentially expressed in astrocyte cluster 6, “astrocyte inflamed in MS”. These astrocytes were shown to have multiple interactions with inflamed microglia within MS [18].

Whereas *SPARC* was only detected in the subcluster of inflamed astrocytes, *SPARCL1* was significantly and differentially expressed in other cell subclusters, included an unclassified immune cluster (0.97 log fold and adjusted *p*-value 1.85 × 10^−42^) and was within an oligodendrocyte group termed “immunological OPCs” (0.41 log fold change, adjusted *p*-value 6.22 × 10^−15^). These OPCs upregulated immunological markers and were frequent in the core of the lesion. These OPCs also upregulated the chondroitin sulfate proteoglycan *VCAN* (0.72 log fold change, adjusted *p*-value 262 × 10^−37^) and *BCAN* (0.47 log fold change, adjusted *p*-value 4.06 × 10^−18^).

In keeping with the divergent expression patterns between *SPARC/SPARCL1* versus SPOCK family members, *SPOCK1-3* expression was detected in different cell subsets. *SPOCK1* was differentially expressed in cluster 7 “perinodal astrocytes” (1.2 log fold, adjusted *p*-value 2.7 × 10^−150^) and cluster 1 “reactive/stressed astrocytes” (1.09 log fold, adjusted *p*-value 2.52 × 10^−131^). *SPOCK1* was identified in OPC subset 1 “stressed OPCs” (0.62 log fold change, adjusted *p*-value 4.39 × 10^−19^). Both *SPOCK1* (2.22 log fold, adjusted *p*-value 4.54 × 10^−109^) and *SPOCK3* (1.45 log fold, adjusted *p*-value 9.70 × 10^−37^) were identified in subcluster 6 of vascular cells, thought to represent misclassified oligodendrocytes due to *MOBP* and *PLP1* expression. *SPOCK2* was in the “T cell” immune cluster (1.2 log fold change, adjusted *p*-value 1.60 × 10^−233^) and vascular cell cluster 1, annotated as endothelial cells (0.93 log fold change, adjusted *p*-value 4.49 × 10^−82^). The different expression characteristics between *SPOCKs*, *SPARC*, and *SPARCL1* further highlight their divergent roles and may explain their dichotomous trends in expression across MS lesions. Although their expression was not significantly different across control or MS lesions, *SMOC* members were significant differentially expressed genes in the “immunological OPC” cluster (0.265 log fold change, adjusted *p*-value 6.24 × 10^−10^) as well as “normal white matter OPCs” (0.2644 log fold change, adjusted *p*-value 2.97 × 10^−11^). *SMOC2* was in vascular subset “pericytes” (1.16 log fold change, adjusted *p*-value 1.41 × 10^−65^) and “smooth muscle cells” (1.22 log fold change, adjusted *p*-value 7.80 × 10^−17^).

### 2.5. Immunohistochemistry of SPARC in MS Lesions

To affirm the transcriptome results with protein expression, we stained chronic active MS lesions for the expression of SPARC. We started with immunoperoxidase labeling of a demyelinated lesion devoid of staining for proteolipid protein (PLP) and characterized the lesion as chronic active based on elevated HLA-DR staining in the rim of the lesion (Figure 4). Staining for SPARC shows elevated immunoreactivity in cellular profiles, particularly in the soma of cells (Figure 4) in the lesion center and rim.

Next, we used immunofluorescence microscopy to facilitate the use of several antibodies simultaneously. In a demyelinated chronic active juxtacortical white matter lesion with CD45+ immune cells localized at the edge, SPARC was observed throughout the lesion center and edge but not in the surrounding white matter (Figure 5A,B). SPARC was also observed throughout an active lesion, characterized by a high density of CD45+ cells across the plaque (Figure 5C).

## 3. Discussion

The purpose of our study was to undertake a detailed investigation into transcriptome changes of the ECM across MS lesions. This paper used a classification system of the ECM developed by Naba et al. (2012) [12,14]. Appendix A show the categorization of the ECM. Limitations in our study include a low sample size of MS lesions, which included three MS lesions from three patients in the spatial dataset and lesions from five progressive MS patients from the Absinta dataset. Nonetheless, our results highlight the significant alterations that ECM members undergo during MS, as well as how the ECM differs between the edge and core of chronic active lesions. These findings correlate with known literature, identifying the upregulation of *SPP1* and *VCAN*, both ECM members with detrimental roles in the MS disease course [6,7,17].

*SPP1*, the gene that encodes osteopontin, had the greatest mean UMI count of ECM members in white matter active lesions edges and a downregulated expression in lesion cores, in both spatial and single-cell datasets. Osteopontin has been identified as an upregulated protein in MS, and its expression has been attributed to inflammatory macrophages [17]. Osteopontin also altered microglial activation, decreased inflammatory cytokines in cell culture [19], and was associated with worsened outcomes in a spinal cord injury model, likely due to its ability to switch microglia activation to a harmful phenotype [17].

Our in-house spatial and the publicly available single-nucleus datasets found that the greatest differentially expressed ECM members across MS lesions are matricellular proteins. In particular, *SPARC* and *SPARCL1* were upregulated in both the core and edge of chronic active lesions and had some of the greatest median-normalized expressions of ECM components in both the spatial and the public single-nucleus dataset. We correlated these transcriptome findings with immunohistochemistry in chronic active MS lesions.

Within the Absinta et al. (2021) dataset [18], *SPARC* and *SPARCL1* were significantly differentially expressed genes within the “astrocytes inflamed in MS” subsets. *SPARC* also appeared within a list of lysolecithin-induced genes in astrocytes (1.8 log fold change after 24 h, adjusted *p*-value < 0.05) [20]. *SPARC* and *SPARCL1* transcripts have appeared in other datasets, with higher expression in normal-appearing spinal cord gray matter than white matter and an increase in spinal cord lesions [17]. SPARC and SPARCL1 proteins were expressed by radial glial cells, astrocytes, and developing vessels [21,22]. SPARCL1 was also expressed on mature resting microglia throughout the cortex [23]. Following various injuries, SPARCL1 was downregulated in microglia but upregulated in peri-lesional reactive astrocytes [21,23,24]. Similarly, SPARC was upregulated by reactive astrocytes following denervation in the hippocampus [25] and upregulated near blood vessels following a cortical lesion in mice [26].

Consistent with transcriptome upregulation, the immunohistochemistry of MS lesions revealed that SPARC was elevated in MS lesions. However, future studies are important to further investigate the protein elevation of the SPARC family. As reviewed elsewhere, the ECM can alter MS disease outcomes, and, conversely, the neuroinflammation in MS alters the composition of the ECM [27]. There are many crucial elements still unknown in MS pathogenesis that can alter ECM composition, such as the role of environmental exposures [28]. Finally, this study did not investigate the consequence of SPARC upregulation in MS. The literature on the role of SPARC family in MS is currently limited. One study found SPARCL-1 was differentially expressed by proteomic analysis in cerebrospinal fluid and was higher in secondary progressive MS versus relapsing–remitting MS but could not replicate this finding in other datasets [29]. SPARC and SPARCL1 can be upregulated by astrocytes in inflammatory conditions and likely influence their activity. Further studies will be important to understand their exact functions in MS lesions.

## 4. Materials and Methods

### 4.1. Postmortem Multiple Sclerosis Specimens for Spatial RNA-seq

The Visium spatial gene expression platform by 10x Genomics was used to conduct the spatial transcriptomics analysis. The methods were described previously by Dong et al. [19]. Tissues were from two control subjects, and 3 patients with MS with the following history and lesions: 60-year-old female with progressive MS with a chronic active white matter lesion, 61-year-old male with progressive MS with a chronic active white matter lesion, and 26-year-old male with relapsing–remitting MS with a cortical lesion (overlying meninges and involving both white and gray matter).

Brain sections were first stained with H&E. Following this, permeabilization, reverse transcription, second-strand synthesis and denaturation, cDNA amplification and quality control, library construction, and sequencing were performed according to Spatial Gene Expression User Guide (CG000239). Sections were loaded at 300pM and sequenced on a NovaSeq 6000 System (Illumina, San Diego, CA, USA) using a NovaSeq 200 cycle S1 flow cell. Sequencing was performed using the following read protocol: read 1, 28 cycles; i7 index read, 10 cycles; i5 index read, 10 cycles; and read 2, 90 cycles.

### 4.2. Spatial RNA-seq Analysis

The sequencing depth obtained ranged from 1.20 to 1.42 × 10^6^ reads per library. The base call files, BCL files, and histology images were processed using 10x Genomics Space Ranger v1.2 analysis pipelines, which use STAR v.2.5.1 for genome alignment against the GRCh38 human reference dataset. The Space Ranger Count pipeline incorporates the H&E image and fastq files to perform tissue alignment and UMI counting. The count files generated for each library were then aggregated with normalization set to ‘Mapped’; this mode of normalization subsamples reads from higher-depth capture areas until that all have, on average, an equal number of reads per tissue covered spot that are confidently mapped to the transcriptome). The spatial dataset is publicly available (see ‘Data Availability Statement’ below). The aggregated cloupe file was then visualised in Loupe Browser (10x Genomics). The cloupe file included the H&E stain, which was used to define white matter and gray matter in control tissue and to identify the active edge and inactive core of chronic active lesions and normal-appearing white matter. Active lesion edges were defined histologically as regions with myelin loss and infiltrated immune cells, aided by the immunofluorescence of the original MS lesions (MBP; neurofilament-L/M/H; laminin; CD45; and DAPI). Inactive lesion cores were histologically defined as the hypocellular lesion centre with cellular debris and minimal immune cells. Periplaque white matter borders were defined as white matter adjacent to lesions. The borders of each category were aided by unbiased clustering, as the active lesion edge, lesion core, and periplaque white matter separated into distinct clusters. The list of ECM genes was imported, and Loupe Browser generated heat maps and *p*-adjusted values of differentially expressed genes. *p*-values were adjusted for multiple testing using the Benjamini–Hochberg procedure and expression values are reported as the median-normalization average of a spot, which represents the mean of observed UMI counts normalized by the size factor for each spot in the representative cluster.

### 4.3. Human Single-Nucleus RNA Analysis

Single-nucleus RNA-sequencing analysis (snRNA-seq) of frozen human brain tissue was derived from a dataset provided by [18]. In this study, authors isolated mixed and inactive white matter MS lesions at different stages of inflammation as well as the demyelinated lesion core, the white matter periplaque, and normal white matter from neurologically healthy brains. The dissection of human brain tissue and nuclei extraction have been described in their methods. They included brain tissue blocks from the Netherlands Brain Bank (5 patients with progressive MS and 3 age- and sex-matched non-affected, non-dementia controls). Their methods used the Chromium Single Cell 3′ gene expression platform (10x Genomics). The fastq files were demultiplexed using the 10x Genomics Cell Ranger pipeline. The count files generated for each library were then aggregated with normalization set to ‘Mapped’. The aggregated cloupe file was visualised in Loupe Browser.

In LoupeBrowser (v6.3.0), only nuclei with <5% of mitochondrial contamination and between 200 and 2500 genes expressed were retained. All expression values are reported as the median-normalization average of a gene, which represents the mean of observed UMI counts normalized by the size factor for each cell in the representative cluster. It was calculated through Loupebrowser with the equation mean-normalized average = mean ([gene_X_UMI_count × size_factor for cell in cluster]).

### 4.4. Immunofluorescence Staining

Post-mortem frozen brain tissues from people with MS were obtained from The Multiple Sclerosis and Parkinson’s Tissue Bank situated at Imperial College, London (https://www.imperial.ac.uk/medicine/multiple-sclerosis-and-parkinsons-tissue-bank; accessed on 15 January 2024). This bank has been approved as a Research Tissue Bank by the Wales Research Ethics Committee (Ref. No. 18/WA/0238). Lesions from the brains of two females (aged 50 and 42 years, MS-230 and MS-338) were analyzed for this study. No lesion was detected in MS-297 (female; aged 58 years).

Slides were thawed at room temperature for 15 min, then delipidated via sequential wash with 50%, 70%, 90%, 95%, 100%, 95%, 90%, 70%, and 50% ethanol. Slides were then fixed with 4%PFA for 15 min. Tissue sections were permeabilized with 0.25% Triton X-100 in PBS for 10 min. Blocking of samples was performed using horse serum blocking solution (0.01M PBS, 10% horse serum, 1% bovine serum albumin (BSA), 0.1% cold fish skin gelatin, 0.1% Triton-X100, and 0.05% Tween-20) for 1 h at room temperature. Tissues were then incubated overnight at 4 °C with diluted primary antibodies in antibody dilution buffer (PBS, 1% BSA, 0.1% cold fish stain gelation, and 0.1% Triton X-100). The following primary antibodies were used: chicken anti-human myelin basic protein (MBP) (1:1000; ThermoFisher, Waltham, MA, USA, PA1-10008), rat anti-human CD45 (1:500; Abcam, Waltham, MA, USA, MA5-17687), and mouse anti-human SPARC (1:100; R&D, Minneapolis, MN, USA, MAB941). Next, slides were washed three times, for 5 min each with PBS containing 0.1% Tween-20 and then incubated with the corresponding fluorophore-conjugated secondary antibodies (1:400; Jackson ImmunoResearch Laboratories, West Grove, PA, USA) and 4′,6-diamidino-2-phenylindole (DAPI) (1 µg/mL, Sigma, Burlington, MA, USA) suspended in the antibody dilution buffer for 1 h at room temperature. After washing (3 times 5 min each), slides were mounted using Fluoromount G (Southern Biotech, Birmingham, AL, USA). Isotype controls were included. Images were captured on the Leica TCS SP8 confocal laser scanning microscope and Olympus VS110 Slide scanner.

## 5. Conclusions

In conclusion, our findings are proof of principle that modern sequencing techniques can lead to the identification of novel ECM targets, and they highlight how there is a prominent alteration in a variety of ECM members in MS lesions. We identified SPARC and SPARCL1 as novel ECM members highly upregulated in MS lesions. These results underline the importance of public datasets and that datamining can yield new ECM targets to understand the role of these ECM members within MS. The ECM remains an important modulator in MS pathogenesis, and this study highlights that there are multiple overlooked ECM targets that require further research to understand how they can be controlled to modulate inflammation and repair.

## Figures and Tables

**Figure 1 ijms-25-01240-f001:**
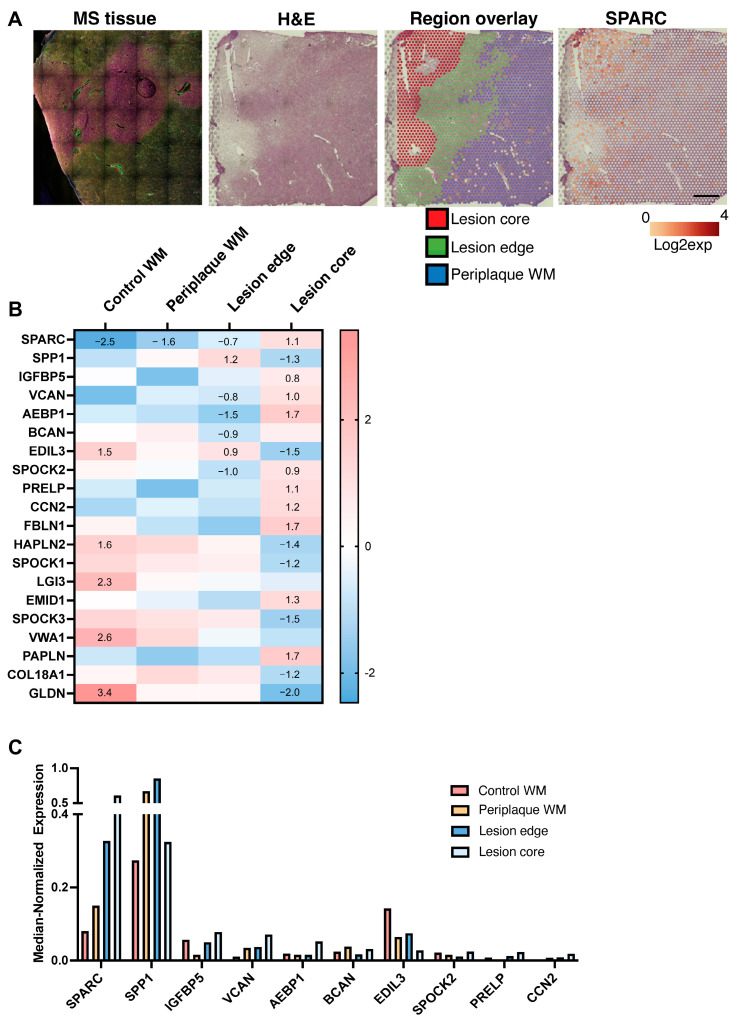
Spatial expression of the ECM in MS lesions. (**A**) Immunofluorescence of original MS lesion (far left) stained with MBP yellow; neurofilament-L/M/H magenta; laminin red; CD45 green; DAPI blue. Representative image of MS spatial lesion, H&E-stained in LoupeBrowser (middle left). Region overlay of lesion core (red), lesion edge (green), and periplaque white matter (blue) are displayed (middle right), and a representative log2 expression of *SPARC* is superimposed on the H&E (far right). Scale bar represents 2.5 mm. (**B**) Heatmap of control WM and WM MS chronic active lesions, with log2 fold changes in ECM in one condition against all other conditions. Genes are ranked in order of greatest (top) median-normalized expression (UMI count per spot) for MS lesion core. Values shown in heatmaps are the significant (adjusted *p*-value < 0.05) log2 fold expression in that condition compared to all other conditions. (**C**) Top 10 significant (adjusted *p*-value < 0.05) differentially expressed ECM members within white matter. WM = white matter.

**Figure 2 ijms-25-01240-f002:**
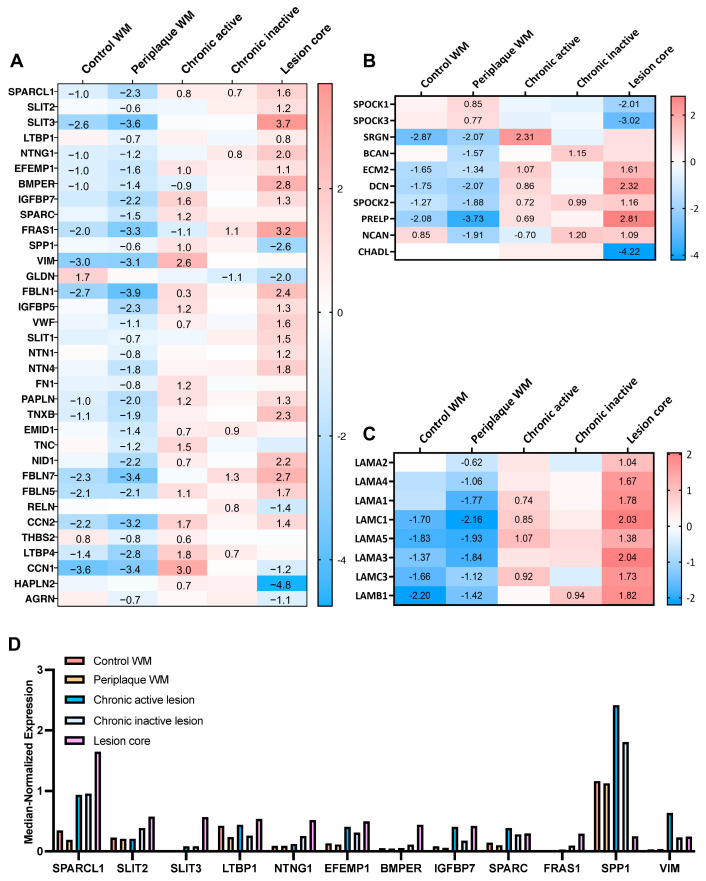
ECM transcriptome changes in different types of MS lesions. Heatmaps of log2 fold differentially expressed genes for one condition compared to all other conditions, ranked in order (top = greatest) of median-normalized expression in lesion cores. Only significant (adjusted *p-*value) log2 fold values are shown for (**A**) major glycoproteins, (**B**) proteoglycans, and (**C**) laminins. Absent values in heatmap represent log2 fold changes that were not significant and show instead the colour value of the log2 fold change. (**D**) Graph of top differentially expressed glycoproteins, in order of greatest expression in lesion cores.

**Figure 3 ijms-25-01240-f003:**
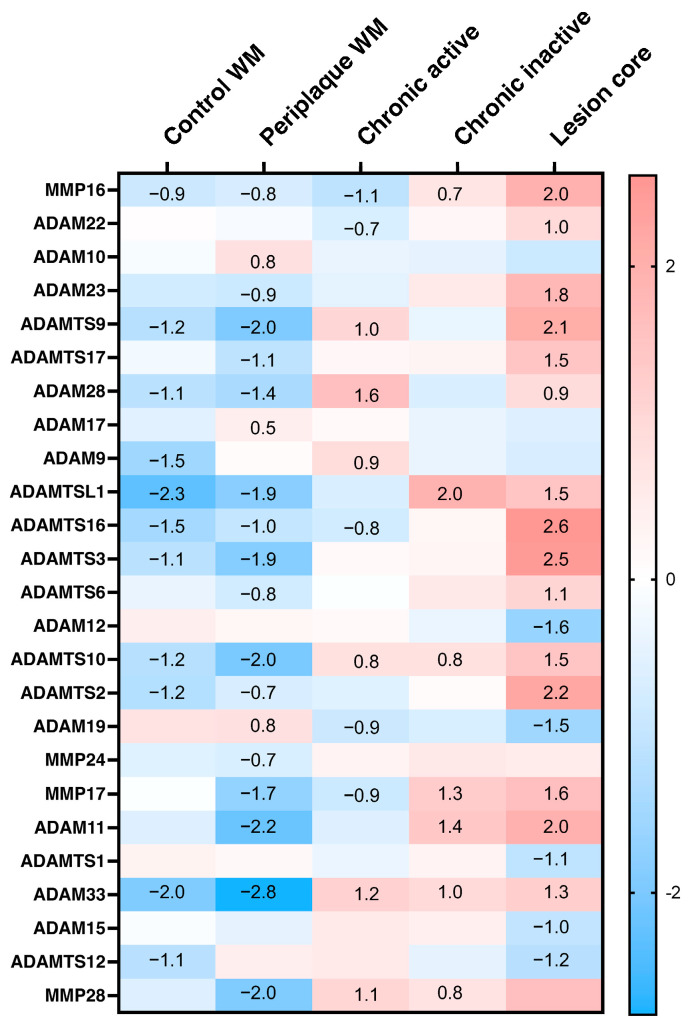
Greatest median-normalized expression of differentially expressed ECM-proteases upregulated per condition compared to all other conditions. Listed genes have adjusted *p*-value < 0.05 log2 fold change and median-normalized expression (normalized UMI counts) greater than 0.05 in at least one condition. Genes are ranked based on greatest mean expression in lesion core (top = greatest), and values in boxes represent significant (adjusted *p*-value < 0.05) log2 fold change compared to other conditions (scale bar on right represents log fold change).

**Figure 4 ijms-25-01240-f004:**
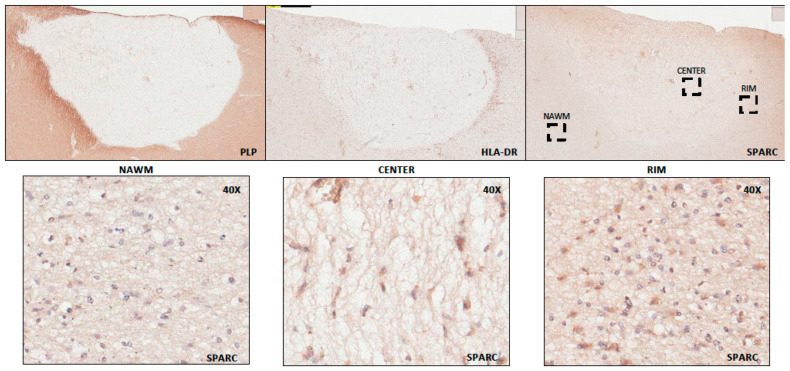
Immunoperoxidase labeling of a chronic active MS lesion shows SPARC labeling that is prominent in the soma of cells in the lesion center and rim. Proteolipid protein (PLP) loss highlights an area of demyelination, while more intense HLA-DR staining in the edge depicts the lesion as chronic active. Higher magnification micrographs are displayed in the lower panels for SPARC in a normal appearing white matter (NAWM), at the lesion center and at the edge. Top images are 2.5× magnification, bottom images are 40× magnification.

**Figure 5 ijms-25-01240-f005:**
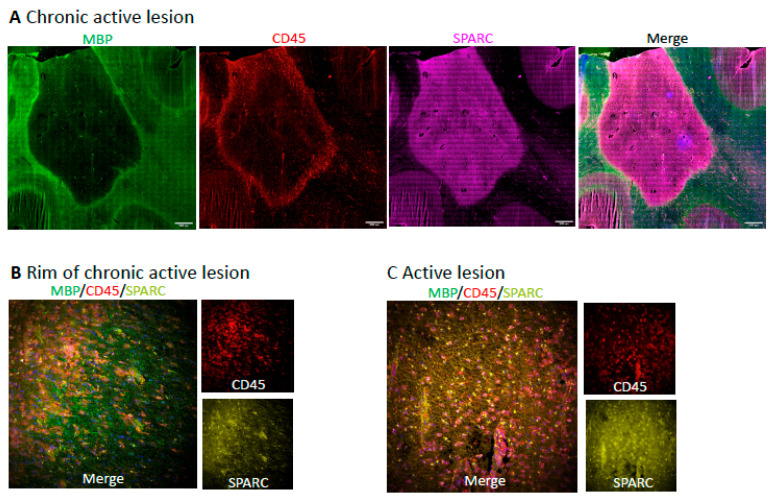
(**A**) Chronic active lesion shows SPARC distributed throughout the lesion but not in adjacent NAWM. Scale bar represents 1 mm. (**B**) SPARC is elevated in the lesion portion (left of frame) of the edge of a chronic active lesion; relative intact MBP profiles can still be observed in the outer edge of the lesion. (**C**) SPARC is prominent across an active lesion. Bottom images are 25× magnification.

## Data Availability

A publicly available dataset was presented in this study and is available as fastq files in GEO (https://www.ncbi.nlm.nih.gov/geo/query/acc.cgi?acc=GSE180759; accessed on 15 January 2024) under accession number GSE180759, PMID: 34497421. The spatial RNA-seq dataset is available to download from the NCBI Sequence Read Archive (SRA) with BioProject accession number PRJNA734097. All other datasets are available upon request.

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
