# Peer review of "Uncovering Novel Extracellular Matrix Transcriptome Alterations in Lesions of Multiple Sclerosis"

_ijms, 2024, doi:10.3390/ijms25021240_

Round 1
Reviewer 1 Report
Comments and Suggestions for Authors
Stephenson et al. provide an integrative analysis of spatial transcriptomic dataset in the context of Multiple Sclerosis (MS). They highlight extracellular matrix (ECM) modifications occurring in the development of the disease, including proteoglycans and glycoproteins alterations. Some of these findings were not described previously, such as the SPARC family, which could be a potential therapeutic target.
The approach of integrating spatial information with RNA-seq data is relatively new and despite being a promising idea, the major limitation of the study is due to the small sample size for both datasets. Furthermore, the number of differentially expressed genes from the reanalysis of the Absinta dataset is very small in size (115 genes), probably because limited to those genes that are part of the core matrisome. I wonder whether other differentially expressed genes can be found in this dataset or other single-cell RNAseq datasets of MS, such as shown in https://www.biorxiv.org/content/10.1101/2022.11.03.514906v1. For example, it would be more convincing whether similar results could be found in similar datasets studying MS in mice (https://elifesciences.org/reviewed-preprints/88414).
To further prove the involvement of the ECM in MS and their related pathways, I think that this study would benefit from an integrative analysis with other sources of data, such as protein-protein interactions (such as the STRING DB) to study the interactions between up- and down-regulated genes, which could form separate molecular modules. I think that a crucial paper that the authors should take inspiration from would be the following:
https://www.nature.com/articles/s41593-022-01097-3
Because the etiology of MS is still unknown, I am curious whether the authors can speculate the origin of some of these ECM alterations from environmental pollutants, such as Bisphenol A (BPA) as observed for other pathological conditions (https://www.sciencedirect.com/science/article/pii/S0269749123013611).
Comments on the Quality of English Language
No comments
Author Response
We appreciate the useful suggestions supplied by all 3 reviewers and hope that we have sufficiently addressed their comments below.
Reviewer 1
Comment: The major limitation of the study is due to the small sample size for both datasets. Furthermore, the number of differentially expressed genes from the reanalysis of the Absinta dataset is very small in size (115 genes), probably because limited to those genes that are part of the core matrisome.
Response: This is an excellent point raised by the reviewer and we agree that because we limited our gene list to the core matrisome, there were fewer genes in our analysis. However, we were intentional in limiting our analysis to the core matrisome since that is our area of focus in this paper. It is also demonstrated in the literature that while ECM members can have lower mRNA expression changes, they persist at the protein level to cause profound consequences. With our focused gene list, we were able to notice interesting trends across the two datasets.
While our gene list was focused, our datasets included an adequate amount of MS lesions which comprised 4 MS lesions from 4 patients in the spatial dataset, and lesions from 5 progressive MS patients from the Absinta dataset. It is outside the scope of this current paper, but our future plans are to interrogate other datasets with more MS lesions to determine whether there are similar matrisome changes and we have listed this future aim in the discussion.
Comment: This study would benefit from an integrative analysis with other sources of data, such as protein-protein interactions (such as the STRING DB) to study the interactions between up- and down-regulated genes, which could form separate molecular modules
Response: We agree that the interaction of the ECM and related pathways of MS is of crucial importance. However, the integration of further analysis such as protein-protein interactions would entail a significant amount of work and is outside the scope of our current paper. We undertook this current study as a preliminary introduction to RNA sequencing alterations in the matrisome and have included in the discussion that this is a limitation, and that we plan future studies to investigate the consequences of these ECM alterations (see Discussion).
Another reason we did not undertake protein-protein pathway investigations is due to the limitations of spatial data. Spatial does not have the resolution of single cell analysis, but are instead looking at groups of cells with the resolution of 50 micron. Thus, each spot could contain multiple cells, and we were cautious not to overinterpret our data to suggest that ECM members are interacting within each spot, or that a single cell was responsible for producing the ECM members. We hope the Reviewer agrees with our rationale.
Comment: I am curious whether the authors can speculate the origin of some of these ECM alterations from environmental pollutants, such as Bisphenol A (BPA)
Response: This is a good observation and we have now added in discussion that “There are likely more key factors in MS pathogenesis, such as the role of environmental exposures that have shown to increase ECM production” (lines 355 and 356) and cited Lamberto et al. 2023.
Reviewer 2 Report
Comments and Suggestions for Authors Alterations to the extracellular matrix (ECM) are a hallmark of MS, but what is lacking is a comprehensive picture of the panoply of changes to ECM component expression in the context of chronic active lesions. To address this problem, the authors first compile an exhaustive list of glycoproteins, proteoglycans and collagen by mining in silico resources. They next conduct spatial RNA-seq from MS vs healthy controls; within MS, they distinguish between normal appearing white matter (NAWM), lesion rim and lesion core. In doing so, they identify a handful of "matrisome" transcripts that are differentially regulated. Next, they reanalyze snRNAseq data from Absinta et al, looking at a number of tissues associated with chronically active lesions including lesion core. Here, they identify an impressive number of major glycoproteins, proteoglycans and laminins that are differentially regulated (and frequently upregulated) in the lesion core. Interestingly, a substantial number of matrix remodelers are also differentially expressed. Finally, the SPARC family of transcripts being increased in expression in both data sets, the authors investigate further by conducting immunoperoxidase and IF staining of chronic active MS lesions. They find that SPARC is expressed in lesion center and rim as opposed to NAWM. Overall, this is a novel and important contribution to our understanding of matrix remodelling in the context of chronic MS lesions. The microscopic analysis of lesions nicely complements the in silico approaches. Nevertheless there are a few issues that need to be addressed prior to publication. These are listed below. It would be helpful to have the lit search strategy (to indentify the list of genes of interest) presented formally in the methods. Lines 114-120: this information may better belong in the Methods. Within the results, it would be more helpful to have some description of the importance of NAWM, lesion rim and lesion core, and their relative significance to disease. These concepts may be familiar to an MS specialist but are less so to a non specialist. There is some confusion with respect to how the data in Fig 1 are described. Ex, in the text there is reference to a Fig 1D which does not exist. In the legend to Fig 1C, there is reference to a heatmap but this subpanel does not contain one. Further, there is some confusionon the part of the reviewer as to what is the definition of "highly expressed" in lesion cores; ex Spp1 seems strongly downregulated in 1B but is the second ranked gene in 1C - is this assessed by normalized reads? Some clarity on these issues would increase reviewer appreciation. Para starting line 156: perhaps best to lead the Fig 1 results with this, as SPARC is the most upregulated? Line 190: the term matrisome is used without prior introductionAuthor Response
We appreciate the useful suggestions supplied by all 3 reviewers and hope that we have sufficiently addressed their comments below.
Reviewer 2
Comment: It would be helpful to have the lit search strategy (to identify the list of genes of interest) presented formally in the methods
Response: We have now formally written the strategy we used to determine which ECM members to include. On line 87, we noted: “We started our investigations by locating a thorough categorization of the ECM. Outside of the core ECM, the list of ECM-associated proteins is difficult to define, since ECM protein domains are shared with non-ECM proteins. Gene ontology categories are of limited help, as they contain lengthy and unannotated lists of proteins. Naba et al., (2012) have expertly curated a thorough categorization of ECM based on protein domains using InterPro and SMART databases (http://smart.embl-heidelberg.de) [14], [16]. Their list of core ECM components correspond with our prior knowledge and expertise of the ECM.”
Comment: Lines 114-120: this information may better belong in the Methods.
Response: We thank the reviewer for this suggestion. The lines were repetitive of what is outlined in the methods (5.2 Spatial RNA-seq analysis) and did not add any extra relevant information. We removed most of the lines and left only “Tissue sections were normalized and mapped using Space Ranger to generate spatially-correlated RNA sequencing mapped to H&E tissue sections. Visualization of data was done using 10x Genomics software Loupe Browser which was used to define borders of normal appearing white matter (NAWM) near lesion, and active rim or inactive core of chronic active lesions.”
Comment: It would be more helpful to have some description of the importance of NAWM, lesion rim and lesion core, and their relative significance to disease.
Response: We have added in descriptions of NAWM, lesion rim, and lesion core, and their histologic features of inflammation, as the level of inflammation in each region is important in MS literature into methods in ‘5.2 Spatial RNA-seq analysis’. “Active lesion rims were defined histologically as regions with myelin loss and infiltrated immune cells. Inactive lesion cores were histologically defined as the hypocellular lesion centre with cellular debris and minimal immune cells. NAWM borders were defined as white matter adjacent to lesions.” We used these designations based on previously published literature by Kuhlmann et al. (Acta Neuropathol (2017) 133:13–24), reference 8.
Comment: Ex, in the text there is reference to a Fig 1D which does not exist.
Response: We have removed this error on line 131 and changed the reference instead to Fig1c. Fig1d was also incorrectly used on line 175 and has now also been corrected to Fig 1b,c.
Comment: The legend to Fig 1C, there is reference to a heatmap but this subpanel does not contain one.
Response: The mention of the heatmap was referring to the panel in 1B. This line has now been moved within the description of 1B.
Comment: there is some confusion the part of the reviewer as to what is the definition of "highly expressed" in lesion cores; ex Spp1 seems strongly downregulated in 1B but is the second ranked gene in 1C - is this assessed by normalized reads?
Response: We appreciate this note by the reviewer for the opportunity to be clearer. Expression was interpreted as UMI counts normalized by the size factor of the dataset.
We have reworded the paragraph (starting at line 171) to more clearly explain that SPP1 was ranked second based on normalized UMI counts while also being downregulated compared to NAWM and lesion rim due to its high overall expression across multiple conditions. We have rephrased to “SPP1, the gene that encodes osteopontin, had the greatest UMI count of ECM members in MS lesion rims. Osteopontin was downregulated in MS lesion cores compared to all other conditions (due to the higher expression in MS lesion rims and NAWM), but still had the second greatest mean UMI count of ECM members in MS lesion cores (Figure 1b,c).”
We have also expanded in the legend for Figure 1 that genes in the heatmap are ranked based on median-normalized expression, which represents UMI count per spot. We have also mentioned that expression represents the mean of observed UMI counts per spot in the beginning of the results, at line 129. For the single cell dataset, we report that expression is represented by UMI count per gene.
Comment: Para starting line 156: perhaps best to lead the Fig 1 results with this, as SPARC is the most upregulated?
Response: This is a good suggestion; we have altered the order of paragraphs so that SPARC is mentioned first.
Comment: Line 190: the term matrisome is used without prior introduction
Response: We have now defined the term matrisome on line 75. (“Beyond the list of core ECM members (proteoglycans, glycoproteins, collagens), there is a much larger group of ECM-associated proteins predicted to be affiliated with the ECM; the combination of these constituents is referred to as the “matrisome” [14]).
Reviewer 3 Report
Comments and Suggestions for Authors
In their paper, Erin Stephenson et al. analyze the matrisome at the mRNA and protein levels in different pathological compartments of MS tissue samples. Although a weakness of this paper is the relatively low number of analyzed samples, the results presented here are of clear interset for our understanding of MS pathophysiology. This is a well-conducted study using state-of-the-art approaches in the field of molecular neuropathology. The article is also very well written. I have only minor comments.
1.The authors indicates in the text that “Figure 1d shows the top 10 differentially expressed genes in control and white matter lesions ranked in order of greatest (top) median-normalized expression in WM lesion cores” but there is no panel d in Figure 1.
2. In figure 1, the authors also chose to perform the following comparisons: “Values shown in heatmaps are the significant (p-adjusted < 0.05) log2 fold expression in that condition compared to all other conditions”. Could the authors show in a data supplement the results obtained when using the control WM as a reference for all comparisons? This could be helpful and would allow nourishing the scientific debate on how normal is the NAWM.
3. I have some reservations regarding the term “lesion rim”used in Figure 1. In light of the results obtained with the LoupeBrowser methodology and considering the fact that this region does not exhibit clear borders and covers a large area as compared to the lesion area, one may wonder if it is actually a lesion rim (lesion edge) or a periplaque region. Lesion edges harbor a high density of myelin-engulfing activated microglia. It would have been interesting to investigate whether this is the case in the regions so-called lesion rims by the authors.
4. The same problems arise from the re-analysis of the results generated by Absinta et al.
- do heatmaps show log2 fold expression in one condition compared to all other conditions?
- the term “lesion edges” was used by Absinta et al. but was not kept by the authors of the present study. I understand that these are semantic issues but this could deserve a litlle explanation in the discussion section.
5. The precise localization of the MS lesion shown in Figure 5 is not indicated in the figure legend. It appears to be a cortical lesion. This type of lesions harbors specificities and this should be kept in mind when interpreting data
Author Response
We appreciate the useful suggestions supplied by all 3 reviewers and hope that we have sufficiently addressed their comments below.
Reviewer 3
Comment: The authors indicates in the text that “Figure 1d shows the top 10 differentially expressed genes in control and white matter lesions ranked in order of greatest (top) median-normalized expression in WM lesion cores” but there is no panel d in Figure 1.
Response: We have removed this error on line 131 and changed the reference instead to Fig1c. Fig1d was also incorrectly used on line 175 and has now also been corrected to Fig 1b,c.
Comment: In figure 1, the authors also chose to perform the following comparisons: “Values shown in heatmaps are the significant (p-adjusted < 0.05) log2 fold expression in that condition compared to all other conditions”. Could the authors show in a data supplement the results obtained when using the control WM as a reference for all comparisons? This could be helpful and would allow nourishing the scientific debate on how normal is the NAWM.
Response: This is an excellent point, and we agree that it is interesting to compare the control and NAWM. However, we did not directly compare control WM against all other conditions in the spatial dataset because of the small sample size (we were limited to 2 control white matter lesions). Nonetheless, to help address your question on differences between control WM and periplaque we did new analysis in the Absinta dataset of the control WM against the periplaque WM and added this in the results. “There were minimal significant changes between periplaque white matter and control white matter, with only CD44 increased in periplaque WM compared to control WM (CD44 log2fold change 1.1 for periplaque versus control white matter; adjusted p-value 0.0001) and four genes sufficiently expressed and significantly decreased in periplaque versus control white matter (i.e., CD74 log2fold change -0.7 with adjusted p-value 0.03; FRAS2 log2fold change -0.8 padj 0.01; THBS2 log2fold change -1.1 padj 2.5E-05; and GLDN log2fold change -1.3 padj 1.35E-08). These results are consistent with the heatmaps in Figure 2, which show similar trends of control and periplaque conditions.”
Comment: I have some reservations regarding the term “lesion rim”used in Figure 1. In light of the results obtained with the LoupeBrowser methodology and considering the fact that this region does not exhibit clear borders and covers a large area as compared to the lesion area, one may wonder if it is actually a lesion rim (lesion edge) or a periplaque region. Lesion edges harbor a high density of myelin-engulfing activated microglia. It would have been interesting to investigate whether this is the case in the regions so-called lesion rims by the authors.
Response: We defined our lesion edge histologically based on the classification of Kuhlmann et al., 2017 (reference 8) as the regions with myelin loss and immune cell infiltration. We had immunofluorescence of the original MS lesions (MBP; neurofilament-L/M/H; laminin; CD45; DAPI) to aid in determining the extent of immune cell infiltration and demyelination. To be clearer for readers, how we classified each region is now listed under methods 5.2 Spatial RNAseq (i.e., “Active lesion rims were defined histologically as regions with myelin loss and infiltrated immune cells. Inactive lesion cores were histologically defined as the hypocellular lesion centre with cellular debris and minimal immune cells”).
The borders of each category were aided by unbiased clustering, as active lesion edge, lesion core, and periplaque white matter separated into distinct clusters.
We also agree with the reviewer that based on Kuhlmann et al., 2017 classification, the white matter near the lesions should be more aptly referred to as periplaque white matter. We have now changed our use of NAWM in the spatial dataset to periplaque WM to correspond to the definition proposed by Kuhlmann et al., 2017.
Comment: Do heatmaps show log2 fold expression in one condition compared to all other conditions?
Response: Yes, they do. Thank you.
Comment: The term “lesion edges” was used by Absinta et al. but was not kept by the authors of the present study. I understand that these are semantic issues but this could deserve a little explanation in the discussion section.
Response: This is an excellent point and we have changed the text to be in line with the original terminology of Absinta et al. The text now refers to chronic active and inactive “lesion edges” rather than lesions. We have also changed the spatial dataset to refer to lesion edges rather than lesion rims so that only one term is used for better consistency across the paper.
Comment: The precise localization of the MS lesion shown in Figure 5 is not indicated in the figure legend. It appears to be a cortical lesion. This type of lesions harbors specificities and this should be kept in mind when interpreting data.
Response: The MS lesion is juxtacortical white matter, does not involve the cortex, and its location is now mentioned in the text (line 303).